# Ti_3_C_2_ MXene Membranes for Gas Separation: Influence of Heat Treatment Conditions on D-Spacing and Surface Functionalization

**DOI:** 10.3390/membranes12101025

**Published:** 2022-10-21

**Authors:** Aline Alencar Emerenciano, Rubens Maribondo do Nascimento, Ana Paula Cysne Barbosa, Ke Ran, Wilhelm Albert Meulenberg, Jesus Gonzalez-Julian

**Affiliations:** 1Materials Science and Engineering Postgraduate Program, UFRN, Natal 59078-570, Brazil; 2Central Facility for Electron Microscopy GFE, RWTH Aachen University, 52074 Aachen, Germany; 3Ernst Ruska-Centre for Microscopy and Spectroscopy with Electrons ER-C, Forschungszentrum Jülich GmbH, 52425 Jülich, Germany; 4Forschungzentrum Jülich GmbH, Institute of Energy and Climate Research, Materials Synthesis and Processing (IEK-1), 52428 Jülich, Germany; 5Jülich Aachen Research Alliance: JARA-Energy, D-52425 Jülich, Germany; 6Inorganic Membranes, Faculty of Science and Technology, University of Twente, 7500 AE Enschede, The Netherlands; 7Department of Ceramics, Institute of Mineral Engineering, RWTH Aachen University, D-52074 Aachen, Germany

**Keywords:** Ti_3_C_2_ MXenes, gas separation membrane, d-spacing control, sieving membrane, H_2_/CO_2_ selectivity

## Abstract

Two-dimensional (2D) MXene materials have recently been the focus of membrane research due to their unique properties, such as their single-atomic-layer thickness, flexibility, molecular filtration abilities and microstructural similarities with graphene, which is currently the most efficient precursor material for gas separation applications. In addition, the potential to process nanoscale channels has motivated investigations of parameters which can improve membrane permeability and selectivity. Interlayer spacing and defects, which are still challenging to control, are among the most crucial parameters for membrane performance. Herein, the effect of heat treatment on the d-spacing of MXene nanosheets and the surface functionalization of nanolayers was shown regarding its impact on the gas diffusion mechanism. The distance of the layers was reduced by a factor of over 10 from 0.345 nm to 0.024 nm, the defects were reduced, and the surface functionalization was maintained upon treatment of the Ti_3_C_2_ membrane at 500 °C under an Ar/H_2_ atmosphere as compared to 80 °C under vacuum. This led to a change from Knudsen diffusion to molecular sieving, as demonstrated by single-gas permeation tests at room temperature. Overall, this work shows a simple and promising way to improve H_2_/CO_2_ selectivity via temperature treatment under a controlled atmosphere.

## 1. Introduction

Regarding the high demand for technologies in the context of sustainability and resource recovery, the development of gas separation membranes has been of major interest in the past few decades [1]. A range of different membranes have been presented to date; however, the necessity for precise and rapid gas separation leads to a trade-off relationship between permeability and selectivity, which stills comprises a major weakness in this field [2]. Therefore, there is considerable demand for membranes with both high permeability and selectivity [3]. The recent use of two-dimensional (2D) materials has been proposed as a promising alternative thanks to simple preparation and the possibility for large scale-fabrication [4]. High-performance two-dimensional (2D) materials, such as graphene oxide (GO) [5], metal-organic framework (MOF) nanosheets [6], covalent organic frameworks (COFs) [7], transition-metal dichalcogenides (TMDSs) [2] and zeolite nanosheets [6,8], have recently been developed for several applications, including in the field of molecular sieving. These 2D materials facilitate selective gas transport by size sieving and/or electrostatic repulsion [2]. The selectivity factor defines the efficiency of a gas separation membrane and is considered ideal when larger molecules are retained while smaller molecules move through the free membrane path. 

In this context, the recently discovered 2D material MXene is an excellent candidate for use in a high-performance sieving membrane (e.g., for H_2_ purification or CO_2_ capture) due to its layered structure, ideal thickness and low-mass barrier [7]. MXenes were first reported by Gogotsi et al. [9]. MXenes consist of transition-metal carbides, nitrides and carbonitrides obtained by selectively etching out the A group (mainly IIIA or IVA), which occupies the interspace in the MAX phase, the precursor material. They are represented by the formula M_n+1_X_n_T_x_
*(n =* 1, 2 or 3), where M is a transition-metal and X is carbon or nitrogen. T represents the nanosheet’s surface functionalization, which appears after centrifugation cycles (O, −OH and −F) and changes depending on the synthesis parameters [10]. The diversity of functional groups leads to the ordered deposition of nanosheets, which can create specific spaces for highly efficient sieving membranes [11]. Membrane thickness is also crucial for the selectivity and high permeability of smaller gases while it maintains the blocking effect of larger molecules [2]. MXene free-standing membranes with different thicknesses have been produced via vacuum-assisted filtration (VAF) [12,13,14,15]. Ding et al. produced a free-standing membrane via VAF with free spacing of 0.35 nm and a thickness of 2 µm, which attained a high H_2_/CO_2_ selectivity (166.6) greater than the Knudsen coefficient (4.7) [14]. The selectivity of H_2_/CO_2_ in an ultrathin continuous 2D CTf-1 membrane (COF, 100 nm) was improved by increasing the membrane thickness while the gas permeance was reduced [11]. 

Nanochannel size plays an important role regarding the parameters which influence the separation performance in terms of permeability and selectivity [16,17]. A few studies have focused on the manipulation of the d-space for improvement in gas separation; for example, Arshadi et al. tailored the free interlayer spacing of the MXene nanosheets through impregnating different types of ions in an electrical separation system [16]. Additionally, chemical tuning of the MXene nanochannel for H_2_ preferential transport was designed to regulate the interlayer space of MXene nanosheets [15]. Cheng et al. post-treated MXenes with a H_2_ atmosphere at higher temperatures to verify if there was a shift in cycling performance; however, the effect of the thermal treatment on interlayer spacing or stacking was not investigated [18]. 

The sintering of MXene films has been reported in a few works, which opened a window to investigate interlayer spacing adjustment. The temperature tolerance of 2D MXene was studied under higher temperatures for hydrogen sieving [19]. The narrowing of interlayer spacing was adjusted via sintering-temperature regulation of MXene films at temperatures up to 500 °C under an air atmosphere [20]. Moreover, previous works have reported the change in free d-spacing to improve properties to be applied in other fields. The intercalation of Ti_3_C_2_, Ti_3_CN and TiNbC MXenes resulted in the increase in c-lattice parameters, resulting in an increase in Li-ion capacity [21]. The study of MXene membranes for gas selectivity is mainly based on the construction of interstitial space and has not been focused on the interaction effect between the termination groups or on the control of interlayer space [2]. 

Although many reports have focused on the size exclusion effect by adjusting the interlayer spacing, some efforts have been made to evaluate the effect of functional groups on gas behavior during diffusion [22]. Studies have reported that the adsorption capacity of CO_2_ in the MXene interlayer is superior due to its higher quadrupole moment compared to other gases (i.e., N_2_), which increases the separation factor for the H_2_/CO_2_ pair by the suppression of CO_2_ into the MXene nanochannels in parallel to the faster diffusion of smaller molecules such as H_2_ [23]. Other works have presented the effect of different functional groups on simulated MD gas permeation for single and mixed gases, and the results show no significant difference between both the permeation behavior of a surface presenting −F and with a variety of functional groups such as −OH, −O and −F [14].

In view of the above information, this work aimed to study the effect of heat treatment under a controlled H_2_/Ar atmosphere and vacuum on the interlayer spacing of MXene nanosheets, as well as the nature of functional groups to improve the membrane’s gas permeation behavior. Thus, MXene nanosheets were produced by the top-down conventional method and Al_2_O_3_ substrates were coated with MXenes via dip coating to effectively produce thin films. The intended thickness was achieved by controlling the Ti_3_C_2_ suspension concentration, and the d-spacing was narrowed through heat treatment at 500 °C under a H_2_ atmosphere to improve the H_2_/CO_2_ selectivity. Single-gas permeation tests were conducted under different pressures at room temperature, and the gas diffusion mechanism was associated with the free path between the nanosheets achieved using different heat treatment conditions of the coated substrate. 

## 2. Materials and Methods

### 2.1. Ti_3_C_2_ MXene Synthesis

The precursor powder for the synthesis of Ti_3_C_2_ MXenes was a MAX phase (Ti_3_AlC_2_), which was produced in-house with a high purity of 98% by mixing Ti, Al and C powders in stoichiometric quantities with KBr powder, followed by pressing the compacted powder and a heat treatment at 1200 °C through the molten salt method as described in [24]. Using the top-down method [25], the Ti_3_C_2_T_x_ MXene was synthetized using HF in situ from the reaction of 1 g of lithium fluoride powder, −300 mesh (Sigma Aldrich, Germany), LiF and 10 mL of 9 M HCl, which was obtained by diluting 10.2 M hydrochloric acid 32% (Sigma Aldrich, Germany) in deionized water to achieve the desired concentration. This reaction was performed in a 50 mL PTFE reaction flask under stirring using a magnetic bar with a rotation speed of 500 rpm for 30 min at room temperature. 

Next, 0.5 g of Ti_3_AlC_2_ (the precursor MAX phase) was introduced stepwise into the acid solution for 5 min. After 24 h at 35 °C of etching aluminum from the precursor, the acid solution was carefully removed and washed with deionized water (DI) by centrifugation. Before each centrifugation cycle, 250 mL of DI water was mixed with the remaining slurry followed by gently shaking the 50mL falcon tubes and subsequent centrifugation at 3500 rpm for 5 min. A pH of 6–7 was achieved after 3 centrifugation cycles. The obtained slurry and supernatant were then separated with a pipette. The slurry, which contained MXene and non-etched MAX phase, was diluted in 100 mL of DI water and sonicated for 1 h in order to enhance delamination of Ti_3_C_2_T_x_ particles and to obtain a high amount of single MXene flakes in the suspension. Surfactants were not needed for the delamination, as MXenes can be well-dispersed in water [21,26,27]. The final suspension after the sonication step was purified through decantation, and so it was left to rest overnight prior to decantation, and the etched and non-etched MAX phases were mechanically separated.

### 2.2. Membrane Processing

Ti_3_C_2_T_x_ membranes with different thicknesses were produced by dip coating using an α-Al_2_O_3_ substrate. Thus, 2 concentrations were studied: $C_{1}$, is the concentration of the final suspension, which was obtained using the method described in the previous section. The second and lower concentration ($C_{2}=0.5\;C_{1})$ was obtained by diluting 25 mL of the $C_{1}$ suspension in 25 mL of deionized water. Freestanding membranes were produced for each suspension by vacuum-assisted filtration of an amount of 3 mL using Whatman Anodisc inorganic filter membrane substrates (AAO, diameter 47 mm, pore size 0.2 μm). The free-standing membranes were used for posterior characterizations of the termination groups and the thermal stability of the material. Thin membranes were obtained by dip coating using α-Al_2_O_3_ Pervatec substrates (diameter 38 mm, thickness 2.2 mm). Next, 10 membranes were produced from each suspension (C_1_ and C_2_) and heat-treated under two conditions: A) at 80 °C, under vacuum, for 24 h (C_1__80 and C_2__80); and B) at 500 °C, under a H_2_/Ar atmosphere, for 6 h (C_1__500 and C_2__500). Finally, five similar repetitions were produced for each of these four samples. 

### 2.3. Permeation Tests

All coated α-Al_2_O_3_ substrates were tested through single-gas permeation under different conditions using He, H_2_, CO_2_, N_2_ and CH_4_. The behavior of the membranes was analyzed under 1, 2, 3, 4 and 5 bar at room temperature. Permeance values were obtained, and the H_2_/CO_2_ selectivity factor was analyzed.

### 2.4. Characterization

The crystal structure from the Ti_3_AlC_2_ MAX phase precursor powder to Ti_3_C_2_ MXene thin film was analyzed using X-ray diffraction, which was performed on a D4 Endeavor X-ray powder diffractometer (Bruker, Germany) using a Cu-K alpha wavelength. The data were collected in the 2Theta range of 5–80° with a 0.02° step size and 0.75 s/step. The d-spacing between the MXene layers of the membranes produced by dip coating was estimated through Bragg’s law [14]. Then, Raman spectra were collected using a InVia Raman Microscope (Renishaw, Hoffman Estates, IL) equipped with a solid-state excitation laser (532 nm) and 2400 lines per mm grating in order to characterize the surface functionalization of MXene layers.. Regions of interest were then characterized by Scanning electron microscopy (SEM) and imaged using a Zeiss Ultra 55 microscope (Zeiss, Oberkochen, Germany). TGA was also performed to investigate the thermal stability of the membrane under different atmospheres and temperatures to analyze the effect of temperature and a controlled atmosphere on the nature of surface functionalization and its implication on H_2_/CO_2_ selectivity. The TGA was performed in a NETZSCH STA 449F3 high DTA (NETZSCH-Gerätebau GmbH, Seligenstadt, Germany) furnace at a temperature range from 25 °C to 1200 °C with a heating rate of 5 °C/min under vacuum and a NETZSCH, STA 449F1, Jupter (NETZSCH-Gerätebau GmbH, Seligenstadt, Germany), at a temperature range from 30 °C to 1100 °C with a rate of 5 °C/min under an Ar and H_2_ atmosphere. 

Transmission electron microscopy was subsequently performed to investigate the order, quality and nanosheet spacing of deposited nanofilms. TEM specimens were cut from the MXenes on the substrate via focused ion beam (FIB) milling using an FEI Strata400 system (FEI Company, Hillsboro, USA) with a Ga ion beam. Further thinning and cleaning were performed with an Ar ion beam in a Fischione Nanomill 1040 (E.A. Fischione Instruments, Inc., Export, USA) at 900 eV and 500 eV beam energy, respectively. TEM imaging was performed by an FEI Tecnai F20 microscope (FEI Company, Hillsboro, USA) at 200 kV. High-resolution high-angle annular dark-field (HAADF) imaging and energy-dispersive X-ray spectroscopy (EDXS) were conducted with an FEI Titan G2 80–200 ChemiSTEM microscope (FEI Company, Hillsboro, USA) equipped with an XFEG, a probe Cs corrector and a super-X EDXS system. The gas permeation performance was studied by analyzing single-gas permeation by flowing the following gases: He, H_2_, CO_2_, N_2_ and CH_4_. A custom massive metal module was employed for this purpose, in which internal space dip-coated Al_2_O_3_ substrates were adjusted with thin metal rings and sealed with rubber rings. The measurements were conducted using a single-gas test rig for porous membranes, GMP1. The flow rate and permeance were obtained in real time through the LabVIEW software, 2016 (National Instruments, Austin/US), for further calculations of permeability and selectivity.

## 3. Results

### 3.1. MXenes Synthesis and Membrane Processing

Figure 1 shows the X-ray diffraction measurement results. The spectrum of the slurry contained several peaks, which correspond to non-etched MAX phases and MXenes. A high amount of Ti_3_C_2_T_x_ phase emerged in the slurry and suspension as a result of Al etching followed by washing cycles, with the main peak (002) appearing between 6° and 8° [25,28]. Aluminum, which was present between the Ti-C layers, was efficiently removed and was not present in the membranes, as observed by the absence of the corresponding MAX phase peaks. The formation of the MXene phase was also confirmed by the dark-greenish color of the supernatant, which came out after the third centrifugation cycle instead of a purple/magenta color, which indicates the presence of secondary phases such as Ti_2_CT_x_ [25] (Appendix A). The small fraction of the precursor MAX phase, which can be observed in the slurry and suspension spectra, disappeared in those representing the Ti_3_C_2_ due the decantation step introduced after centrifugation, which led to the suspension being purified before the membrane processing.

The slurry and the free-standing membrane produced from the vacuum-filtrated supernatant after the third centrifugation cycle are shown in Figure 2. Panels (A) and (B) show the presence of MXene particles in the slurry and indicate the presence of MXene flakes instead of non-etched Ti_3_AlC_2_ MAX phases, which agrees with the XRD results and indicates the efficiency of the synthesis method. It is important to consider that MXene flakes presented a barely distinguishable compact structure from that of MAX-phase powder thanks to the narrow space between the two-dimensional layers, as was also found in Ti_3_AlC_2_ particles. The expanded accordion-like structure usually occurs due the exothermic reaction between HF with a high concentration and Al between Ti-C bonds [9], which was not similar to the MXene microstructure obtained in this work. The observed structure is similar to those of MXenes synthetized at low HF concentrations [25]. It was possible to not only observe MXene flakes, but also the presence of transparent monolayers of Ti_3_C_2_, which can be explained by the easy delamination process which occurs along the centrifugation. The remaining solvated Li ions stayed between the MXene layers after synthesis during this phase [29]. Panels (C) and (D) show the microstructure of the free-standing membrane resulting from vacuum-assisted filtration of 30 mL of the supernatant after the third centrifugation cycle (0.22 mg/mL). The film surface containing deposited Ti_3_C_2_ nanosheets with a thickness of around 280 nm is shown in panel (D). It can be noted that the MXene layers are not fully aligned in a few regions, and there are also some wrinkles. Wrinkles lead to imperfections in the membrane microstructure and lower efficiency in permeation [30]. This factor can be explained by the heterogeneous distribution of terminations [31,32] and the presence of water [33]. Thus, the piled free-standing membrane, which was produced for posterior characterization, showed purity, well-deposited MXene layers and mechanical stability, as observed in previous reports [34], which can reproduce the same quality of those nanosheets deposited on α-Al_2_O_3_ substrates.

Figure 3 and Figure 4 show the spectrum of dip-coated Al_2_O_3_ substrates heat-treated at 80 **°C** under vacuum and at 500 **°C** under a H_2_/Ar atmosphere. The respective graphs comprise the superposition of peaks from samples treated under the same conditions. The graphs indicate the formation of pure MXene phase, where the highest peak (002) showed a sharpness which is characteristic of the stacking of Ti_3_C_2_ nanosheets due to ordered deposition (i.e., by horizontal stacking) [16]. The Al_2_O_3_ phase of the substrates is represented by the peaks marked with red squares and could be observed in all spectra. In this work, the heat treatment at higher temperatures for both samples dip-coated with the two different concentrated suspensions (C_1_ and C_2_) resulted in a shift in the peak from lower to higher angles. A small change in the second and third Ti_3_C_2_ peaks representing samples thermally treated at 80 °C can be noticed when compared to those treated at 500 **°C**. The spectrum of the sample treated at a higher temperature showed a small new peak at 17° [16], which suggests stacking of nanolayers, as also reported by Arshadi et al. When comparing the figures, it is important to note that the main phase was kept even after heat treatment B, which evidently suggests the absence of secondary and unlike phases. Additionally, the superposition of the peaks indicates reproducibility regarding the presence of the Ti_3_C_2_ phase. 

The surface morphology and fracture surface of Ti_3_C_2_ supported on α-Al_2_O_3_ substrates are shown in Figure 5 and Figure 6, respectively. The first sample produced by dip coating (C_1__80) presented a surface microstructure without impurities or defects (Figure 5A), and although Ti_3_C_2_ nanosheets are transparent, the film was thick enough that the microstructure of the substrate could not be seen when compared to thinner membranes (i.e., C_2__80, Figure 6C). Due to the ease of peeling off the substrate, all the samples produced with C_1_ and C_2_ concentrations could provide the fracture surface of the free-standing membrane to be analyzed by SEM. 

The heat treatment had a considerable impact on the membrane thickness produced by different concentrations of suspensions C_1_ and C_2_. A decrease in thickness could be observed with the increase in temperature concerning the heat treatment followed by the membrane processing (i.e., dip coating). MXenes supported on α-alumina substrates treated at a higher temperature (condition B, Figure 5D and Figure 6D) exhibited an almost 20% reduced thickness compared to those treated at 80 **°C** under vacuum (condition A, Figure 5B and Figure 6B). Confirming the narrowing of interlayer spacing demonstrated by XRD in the previous section, the decrease in thickness occurred as expected for those samples treated at 500 **°C** under a H_2_/Ar atmosphere. Additionally, the absence of defects on the surface of those samples heat-treated at lower temperatures was also seen in those sintered at a higher temperature, which shows that the chosen free-oxygen atmosphere is better than the air atmosphere used in previous works for reducing d-spacing due to the higher temperature that led to the presence of secondary phases and defects [20]. 

Figure 7 shows an enlarged view of the region which corresponds to the main peak of MXenes (002), highlighting the shift in the peak (002) towards a higher angle in samples treated at higher temperatures (condition B). Peaks at lower angles, approximately 6.5 °C, correspond to C_1_ and C_2_ suspensions treated at 80 °C (condition A). The peaks which appeared around 8.65 °C represent the main Ti_3_C_2_ peaks of those treated at a higher temperature (500 °C). These observations indicate smaller MXene d-spacing for samples dried under condition B according to Bragg’s law. 

Previous studies have shown a shift in the mean peak (002) through different techniques for controlling interlayer spacing [16,35,36]. Cheng et al. treated MXene with a H_2_ atmosphere at higher temperatures, up to 400 **°C**, to check if there was a shift in cycling performance; however, they did not investigate the effect of the thermal treatment on interlayer spacing or stacking [37]. The magnification of the main peak showed the effect of the temperature. A considerable shift in the peak was achieved for the two different heat treatments, and even the samples presenting different thicknesses (i.e., C_1__80 and C_2__80 or C_1__500 or C_2__500) showed similar (002) peak values when heat-treated at the same conditions. As the diffraction peaks give rise to the d-spacing according to Bragg’s law, the free path can be assumed considering MXene nanolayers are ~1 nm [25]. Such similarities in peak values suggest that temperature under an oxygen-free atmosphere is the main parameter regulating the free path, and does not depend on the membrane thickness. The narrowing of the interlayer spacing can be attributed to water loss and de-functionalization (−OH), and the low free energy at higher temperatures favored the decrease in interlayer spacing [19]. The interaction between the adjacent nanosheet favored changing the −OH part to −O functional groups, which helps to enhance the chemical bonding and consequently to decrease the interlay spacing, as also observed in [37].

Table 1 shows the obtained c-lattice values for the C_1__80, C_1__500, C_2__80 and C_2__500 samples, and their respective free d-spacings are exposed, which suggests similarity between the interlayer space of samples thermally treated at the same temperature. Similarly, the d-spacing in previous reports was obtained through Bragg’s Law by the use of constant c-lattice values. The average (002) reflection at 2θ = 6.55 previously shown in Figure 4 corresponds to an interlayer distance of 0.347 and 0.352 nm for C_1__80 and C_2__80 samples, respectively. The average (002) reflection at 2θ = 8.65 corresponds to an interlayer distance of 0.024 and 0.040 nm for C_1__500 and C_2__500, respectively.

The Raman spectrum was collected from two 1-micron MXene films, each of which was treated at different conditions. The technique was able to provide information about the surface chemistry, stacking and quality of Ti_3_C_2_T_x_. Figure 8 shows the comparison between both the Raman shift, which exhibited characteristic wave numbers correlated to vibrations of Ti_3_C_2_T_x_, and their respective surface functionalization. Similarly, as shown in [38], these vibrations of Ti_3_C_2_T_x_ consisted of E_g_ (in-plane) and A_1g_ (out-of-plane) peaks, both of Ti and C atoms. Thus, the spectrum was divided into three regions: the flake region, which corresponds to a group vibration of carbon, two titanium layers and functional groups; the T_x_ region, which represents vibrations of surface groups; and the carbon region (C region), in which both in-plane and out-of-plane vibrations can be found. Both studied free-standing membranes presented vibrations which correspond to the peaks at 205, 255, 286, 404, 580, 622 and 725 cm^−^^1^, but small differences in peak intensities can be observed. As the laser utilized for this measurement was 532 nm, the first resonant peak coupled with the plasmonic peak was not detected [32]. 

Next, the A1g peak at 205 cm^−^^1^ and the E_G_ peak at 255, 286 and 404 cm^−^^1^ indicate vibrations due to surface groups of titanium. The E_g_ and A_g1_ peaks at 580/622 cm^−^^1^ and 725 cm^−^^1^, respectively, correspond to the carbon vibrations [31]. The line broadening and merging in the spectra is indicative of exfoliation and delamination of the Ti_3_AlC_2_ precursor. Annealing at a higher temperature (condition B) led to the partial removal of functional groups [39]. In agreement with this, a small difference in the general peaks was observed, but the nature of the functional group was maintained. In addition, heat treatment can lead to changes in the surface functionalization of MXenes. There is a tendency for the −OH functional groups to change to −O, which leads to increased membrane mechanical stability and a decrease in d-spacing [32]. Therefore, sintering of the MXene membrane at a higher temperature, such as 500 °C under an oxygen-free atmosphere, led to conservation of the surface functionalization and showed that MXene flakes were not affected by this increase in temperature (i.e., no important changes were noted), which is in contrast to the considerable removal of functional groups after treatment around 500 °C in air [40]. Membranes after heat treatment and consequently after removing water between the nanolayers are supposed to remain chemically stable, leading to longer times when stored in an oxygen-free atmosphere [13,41]; however, they have limitations regarding their use in humid environments, especially at high temperatures [22,42].

The TEM results suggest a distinct deposition of Ti_3_C_2_ nanosheets on the surface of Al_2_O_3_ substrates, which were processed by dip coating using the most concentrated suspension, which was identified as C_1_. Although both of them presented a continued thin film on the surface of a porous Al_2_O_3_ substrate, a slight difference in thickness can be observed between the sample heat-treated in condition A (Figure 9A) and the one heat-treated in condition B (Figure 9C); this agrees with what was seen in SEM images (Figure 5 and Figure 6) resulting from the decrease in the average d-spacing confirmed by XRD analysis (Table 2). However, some defects remained after heat treatment. 

For this work, heat treatment at higher temperatures led to stacking of MXene nanosheets (Figure 9D), while the heat treatment at lower temperatures showed the considerable presence of defects such as an increase in d-spacing between the layers [38,43] due the non-well deposition of MXene nanosheets or the presence of water (Figure 9B). Similar to a study by Zheng, Z. et al. that investigated the surface morphology of MXenes supported on alumina substrates after hydrogen annealing, the nanolayers were well-dispersed in the film; therefore, the boundaries of MXene sheets were more indistinct after hydrogen annealing, which can be explained by the formation of a tightly packed network via the partial sintering of sheets at higher temperatures [43]. 

In order to evaluate the chemical stability of the membrane for further investigations on the heat treatment to narrow the d-spacing between the 2D nanolayers, a TGA investigation was carried out under two atmosphere conditions, namely vacuum and an Ar/H_2_ atmosphere (Figure 10a,b), respectively. The mass loss curve of free-standing MXenes measured under vacuum showed that the slight initial water loss of 4.96% was followed by a mass gain path, indicating oxidation (i.e., conversion of titanium carbide into titanium dioxide), as indicated by the XRD pattern of the final powder. The functional groups between the MXene layers, generally −OH, −F and −O, favor the interaction of the oxygen present in the air or water and promote its oxidation [13], which indicates the presence of remaining oxygen during the measurement under vacuum. 

The chemical stability of MXenes depends on its exposure to an oxidative environment and limits their potential in applications where the longevity of the material is required [44]. Herein, the total loss of water occurred until the temperature achieved a value around 450 °C. Other reports have also shown that water was removed at temperatures higher than 300 °C [45]. Figure 10a indicates an improvement in the chemical stability of the free-standing membrane concerning the thermal evaluation under a protected atmosphere. Similarly, as previously indicated, the first mass loss of 1.58% occurred at temperatures below 200 °C. The second and largest loss in mass, 9.64%, started at a temperature near 835 °C and showed phase transition from Ti_3_C_2_T_x_ to TiC, which took place at higher temperatures when compared to those heat-treated under vacuum. In previous studies, it was observed that the structural phase transition from Ti_3_C_2_T_x_ to cubic TiC may occur at annealing temperatures higher than 900 °C [40], as also observed in this work. The absence of titanium oxide indicates a free-oxygen atmosphere as a proper condition to sinter MXene films in order to achieve smaller free interlayer spacings. 

### 3.2. Gas Permeation Tests

Single-gas permeation tests were carried out at room temperature and under different pressures for all samples, respecting the ascending order of the gas ratio from the smallest to the largest: He, H_2_, CO_2_, N_2_ and CH_4_. Three samples for each condition were reproduced and tested by single-gas permeation. The dip-coated substrates were put into a massive permeation module and sealed with a rubber ring (Appendix A) to measure the gas permeation performance. This work showed a higher permeability of small molecules such as He and H_2_ when compared with lager molecules (e.g., CO_2_, N_2_ and CH_4_). Stacked membranes with a free interlayer spacing of 0.024 nm that were obtained after heat treatment presented low permeability values for smaller and larger molecules due the presence of defects, as previously analyzed by TEM. Only membranes produced by dip coating could be tested due to their mechanical stability when deposited on alpha-Al_2_O_3_ substrates, which held the membrane when a negative pressure was applied (common procedure during the gas switching). 

In order to analyze the influence of the thickness of MXene films on the membrane behavior, samples produced using different concentrations (C_1_ and C_2_) followed by the same drying condition (A or B) were compared, as shown in Figure 11, Figure 12, Figure 13 and Figure 14. Figure 11 and Figure 13 show the permeance curves and H_2_/CO_2_ selectivity of the samples C_1_ and C_2_ treated at 80 °C (condition A), indicated by curves and bars, respectively. It is possible to note that a decrease in thickness led to higher permeances, as also seen in a previous work [14]. However, the C_1__80, ~180 nm thickness showed H_2_ permeance higher than 10 [10³ mol/m²sPa], and the C2_80, ~107 nm thickness increased it to values higher than 250. Similarly, for samples C_1_ and C_2_ annealed at 500 °C (condition B, Figure 12 and Figure 14, respectively), the decrease in thickness was accompanied by higher permeances from approximately >4.6 to >180 [10³ mol/m²sPa]. Additionally, a loose microstructure, which can be seen in Figure 9B, is related to the flux increase and consequently the permeability [46]. 

Jeng et al. showed that the disorder of GO layers induced a short pathway in the membrane, implying the permeation increase. Regarding H_2_/CO_2_ selectivity, slight differences were observed at 1 bar between C_1__80 and C_2__80. Although both presented Knudsen diffusion, which occurs when the mean free path of a gas molecule is larger than the channel through which it travels [47], the thicker membrane with the H_2_/CO_2_ selectivity factor ~4.69 as a dominant gas separation mechanism showed a small increment in selectivity, as selectivity is increased by the decrease in the membrane thickness, as also seen in [48]. In the same way, higher H_2_/CO_2_ selectivity was observed for thicker samples heat-treated at 500 °C, such as C_1__500.

Interlayer spacing and a decrease in defects must be considered as parameters to improve the selectivity among the factors influencing a gas-separating membrane’s behavior [36]. Thus, pure MXene membranes produced using the same concentrated suspension, C_1_, and treated at different temperatures (Figure 11 and Figure 13, respectively), were compared to analyze this important factor. Substrates treated at 80 °C presented an average free-interlayer spacing between ordered nanolayers, approximately 0.35 nm, which enabled molecules such as H_2_ and CO_2_ with different molecular sizes to move due to the large free path. Additionally, the larger spacing previously observed in this work (i.e., defects generated by the non-ordered deposition of nanosheets as well as the presence of water, seen in Figure 9B) strongly influenced the diffusion of molecules with no size selection. In turn, molecular weight defined the selectivity in this diffusion mechanism, which was lower than 4.69 (the substrate in this work was measured and showed a H_2_/CO_2_ selectivity of 4.5, Appendix A). 

Then, a considerable decrease in average spacing was observed after heat treatment at a higher temperature, and as expected, a decrease in flux of all gases and consequently reduced permeance was observed. Additionally, the selectivity increased to 7 (1 bar), which is considered molecular sieving. A free path in the interval between hydrogen and carbon dioxide molecular size should be expected for this diffusion mechanism, which is not in agreement with the average free interlayer spacing obtained in this work, 0.024 nm, constituting well-stacked MXene nanolayers. Thus, the results suggest that larger spacings (defects), which were maintained even after the heat treatment, were capable of selecting larger molecules from smaller molecules, improving the diffusion mechanism. Even though there is a partial removal of functional groups, it is supposed that the decrease in defects and in interlayer spacing was mainly associated with the water removal and functional group modification at high temperature (i.e., −OH tends to change to –O), which contributed to a greater interaction between adjacent nanosheets. In addition, the fact that the adsorption capacity of CO_2_ in the MXene interlayer is superior due to its higher quadrupole moment compared to other gases (i.e., N_2_) explained an increase in the separation factor due to the decrease in CO_2_ diffusion into the MXene nanochannels in parallel with the faster diffusion of smaller molecules such as H_2_ [23]. 

Therefore, this shift in the diffusion mechanism from Knudson to molecular sieving was only observed for thicker membranes treated at higher temperatures due to the instability of the thinner membranes and greater possibility of leakage or mechanical damages (i.e., abrasion) during the module assembly and operation or with the increase in pressure (Appendix A).

Permeabilities and the respective selectivity values for the membrane batches used in this study are summarized in Table 2. As already reported, thinner membranes are required to improve the increase in gas flux and well-defined pore size [49] or interlayer spacing [35] to maximize selectivity. In this work, thinner membranes showed higher permeability values and lower H_2_/CO_2_ selectivity. Among the reasons which explain this behavior, the mechanical instability of the membrane and leakage [49] (Appendix A), which may occur during the single-gas permeation tests and/or preparation of the sample inside the module and/or mechanical instability of the membrane, contributed to the decrease in gas separation efficiency. The small change concerning the slight removal of terminating groups, as observed through the previously discussed Raman spectrometry, might lead to weakening of the chemical interaction between the MXene layers. However, since the removal happened in a smaller proportion, a considerable part of the functional groups as well as their variety was maintained, which contributed to maintaining a strong interaction between the MXene nanosheets. Moreover, the interaction between adjacent nanosheets favored a change of functional groups at higher temperatures (i.e., from −OH to −O terminations), as also observed in [37]. For thinner membranes (i.e., C2_80), the partial removal of the functional groups did not considerably interfere with the selectivity. However, it can be assumed that a lower mechanical instability due to its thinner thickness was the crucial reason for the lower separation performance. Considering long-term stability, the decrease in d-spacing after heat treatment at higher temperatures contributed to stacking MXene nanosheets, which favors chemical stability when compared to individual flakes with a high surface area.

## 4. Conclusions

In this work, pure and flexible MXenes were produced by controlling synthesis parameters, and laminar thin films were processed by dip coating with different concentrations of MXene suspensions. Moreover, the decrease in defects as a consequence of the d-spacing narrowing through heat treatment at 500 °C under an Ar/H_2_ atmosphere promoted a change in the diffusion mechanism, which led to molecular sieving. A heat treatment under an oxygen-free atmosphere was required for layer stacking due to the chemical instability of MXenes. The absence of important changes in the nature of the pristine material as well the conservation of functional groups is an important factor which favored controlling the interlayer spacing through the heat treatment under higher temperatures. In addition, membranes with different thicknesses showed approximately the same interlayer narrowing factor, which led to the assumption that the membrane thickness did not limit the stacking efficiency by heat treatment under a controlled atmosphere. This work indicated a potential method for the efficient manipulation of MXene interlayer spacing by controlling the heat treatment conditions. The good chemical stability of Ti_3_C_2_ flakes after heat treatment at higher temperatures in the presence of H_2_ favors the use of this method to break the permeability–selectivity trade-off of MXene-based membranes for gas separation.

## Figures and Tables

**Figure 1 membranes-12-01025-f001:**
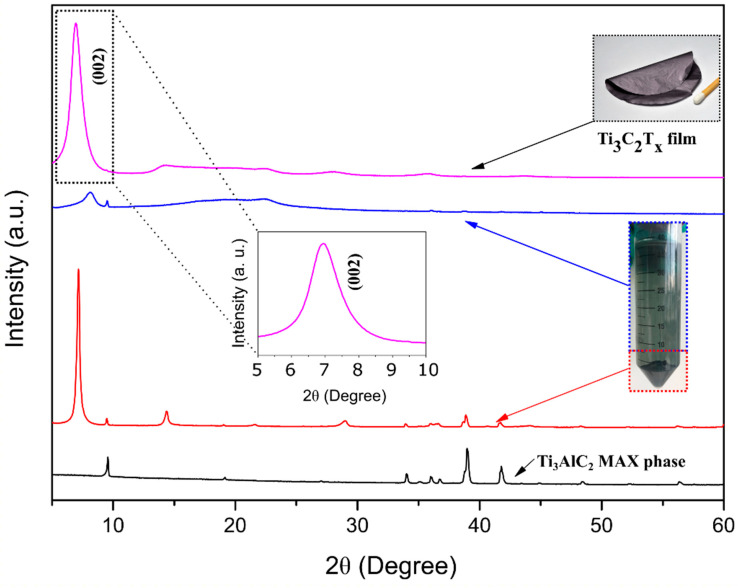
XRD patterns of the MAX (Ti_3_AlC_2_) powder (black), the slurry (red) and suspension (blue) after the third centrifugation cycle and the Ti_3_C_2_ MXene free-standing membrane (pink).

**Figure 2 membranes-12-01025-f002:**
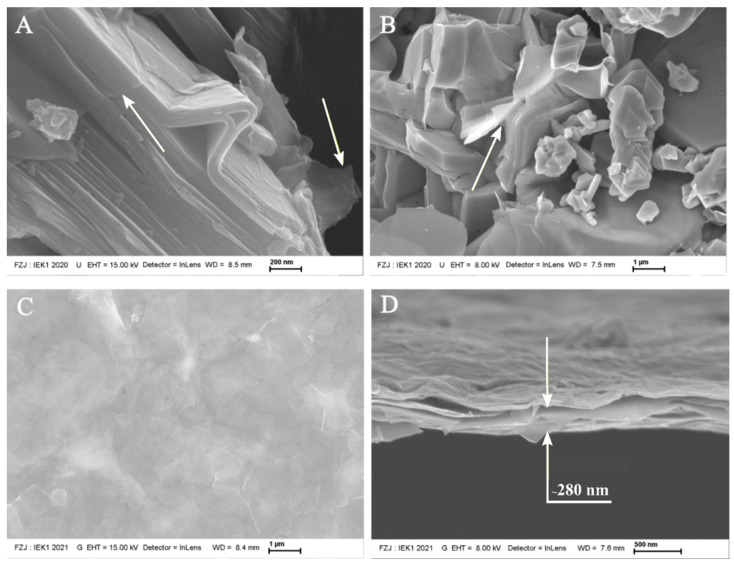
(**A**,**B**): SEM images of the slurry after the third centrifugation cycle showing the presence of transparent nanosheets, as indicated by the yellow arrows. The free-standing membrane: (**C**) surface morphology and (**D**) fracture surface.

**Figure 3 membranes-12-01025-f003:**
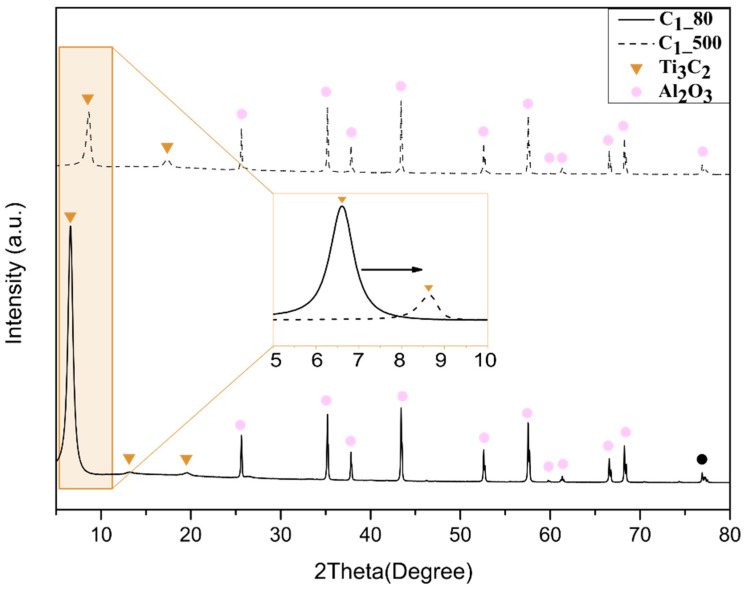
XRD patterns of Al_2_O_3_ substrates coated with Ti_3_C_2_ by dip coating using suspension C_1_ treated at 80 **°C** under vacuum and 500 **°C** under H_2_/Ar atmosphere.

**Figure 4 membranes-12-01025-f004:**
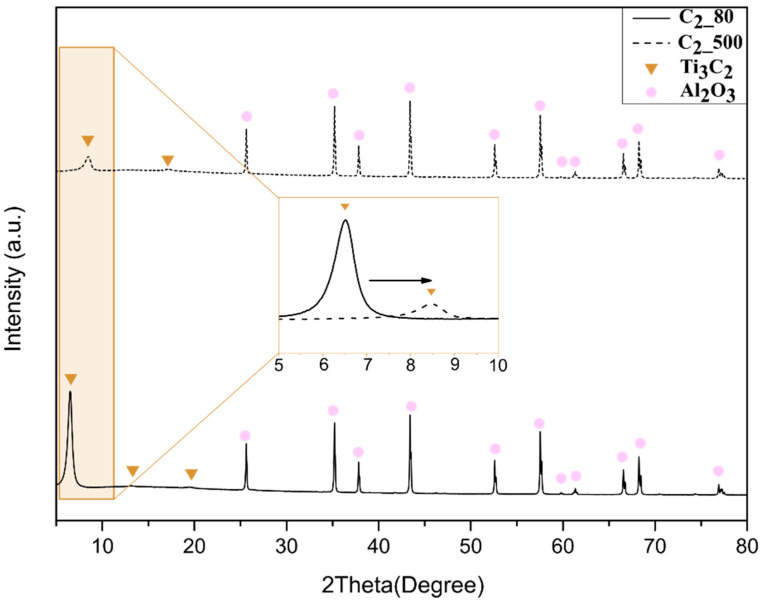
XRD patterns of Al_2_O_3_ substrates coated with Ti_3_C_2_ by dip coating using suspension C_2_ treated at 80 **°C** under vacuum and 500 **°C** under H_2_/Ar atmosphere.

**Figure 5 membranes-12-01025-f005:**
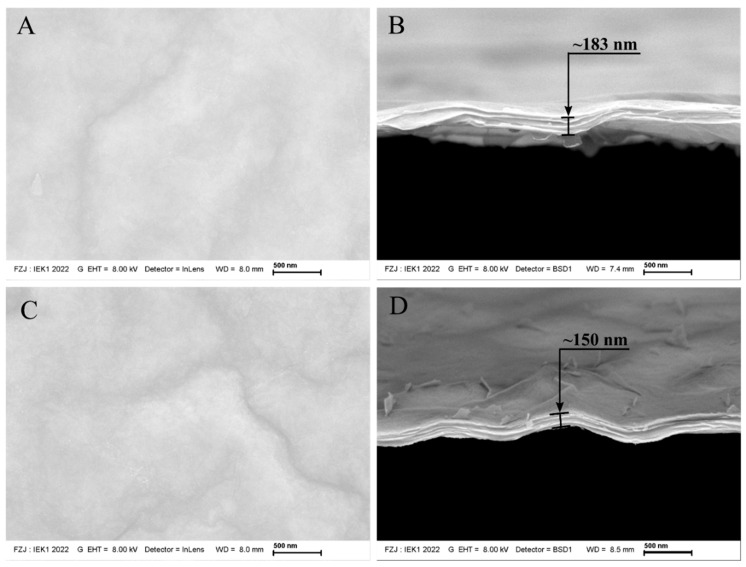
(**A**–**D**): SEM images of Ti_3_C_2_ MXene membranes produced via dip coating on an α-Al_2_O_3_ substrate from suspensions C_1_ treated at 80 °C and 500 °C under H_2_/Ar atmosphere, respectively.

**Figure 6 membranes-12-01025-f006:**
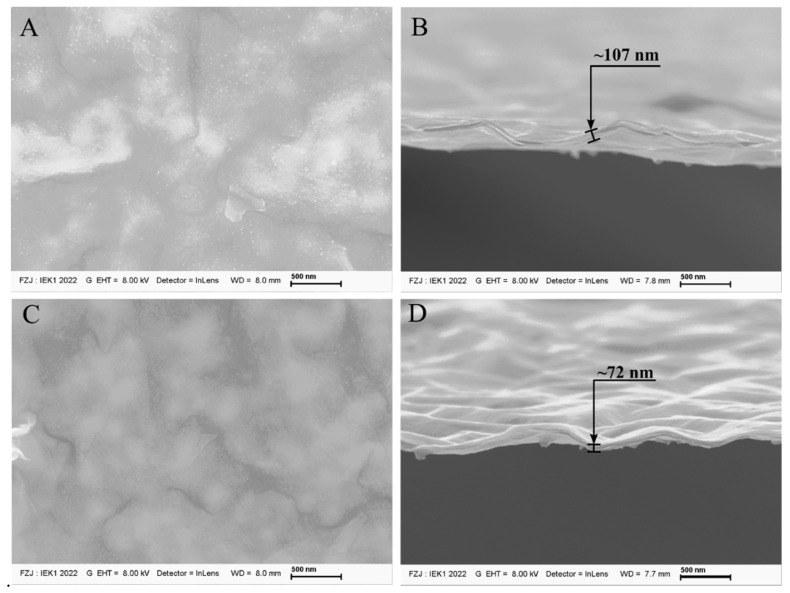
(**A**–**D**): SEM images of Ti_3_C_2_ MXene membranes produced via dip coating on an α-Al_2_O_3_ substrate from suspensions C_2_ treated at 80 °C and 500 °C under H_2_/Ar atmosphere, respectively.

**Figure 7 membranes-12-01025-f007:**
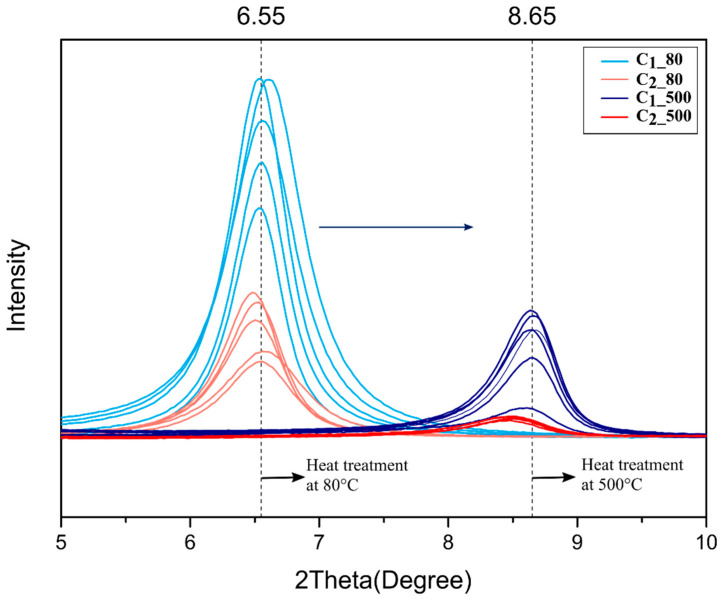
XRD patterns of samples heat-treated at different conditions. An interval from 5° to 10° shows the main MXene peak (002).

**Figure 8 membranes-12-01025-f008:**
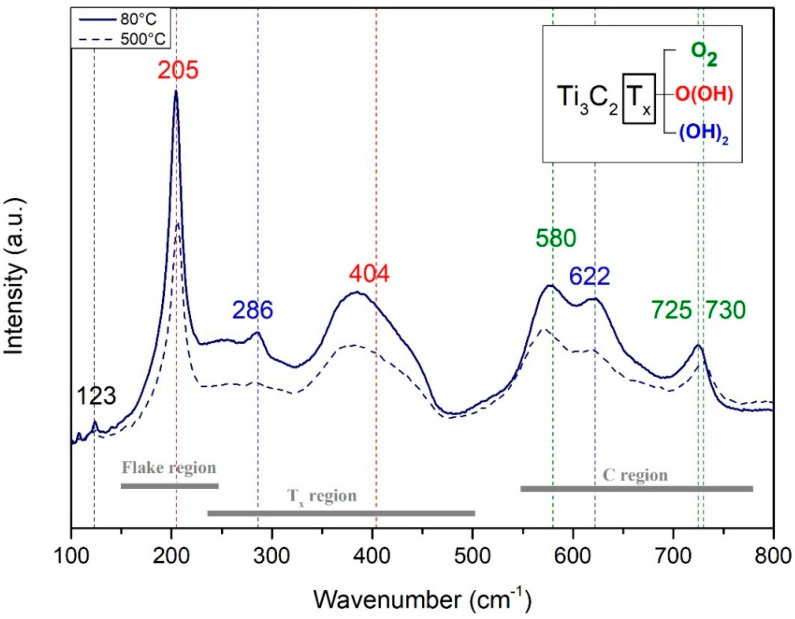
Raman spectra of Ti_3_C_2_ (laser 532 nm). T_x_ functional terminations were compared to the literature.

**Figure 9 membranes-12-01025-f009:**
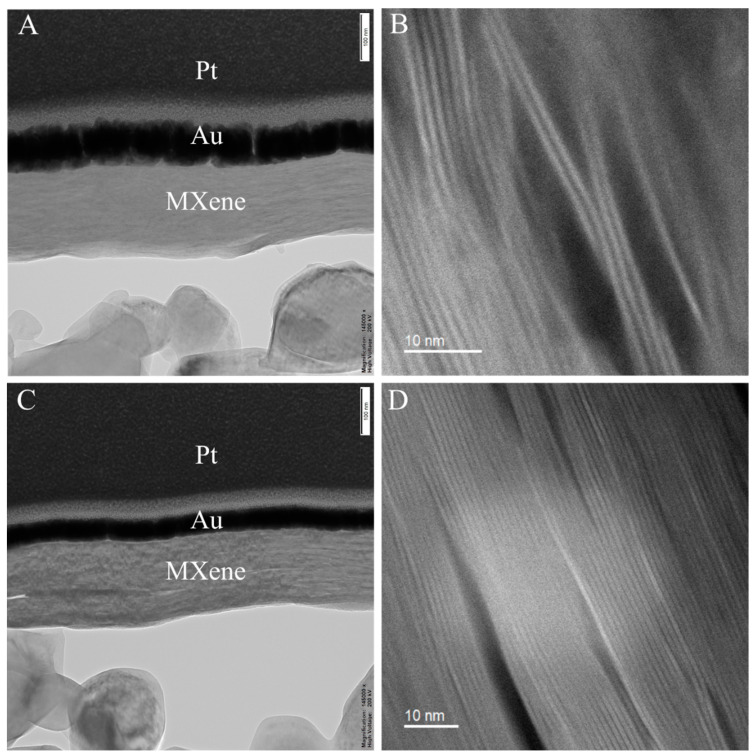
Morphology and structure of exfoliated MXene (Ti_3_C_2_T_x_) nanosheets and stacked MXene membranes. (**A**) SEM image showing the cross-section of the sample C_1__80; (**B**) TEM images of the sample C_1__80 showing regions containing non-ordered and spaced nanosheets; (**C**) SEM image showing the cross-section of the sample C_1__500; (**D**) TEM images of the sample C_1__500 indicating regions with stacked MXene nanosheets.

**Figure 10 membranes-12-01025-f010:**
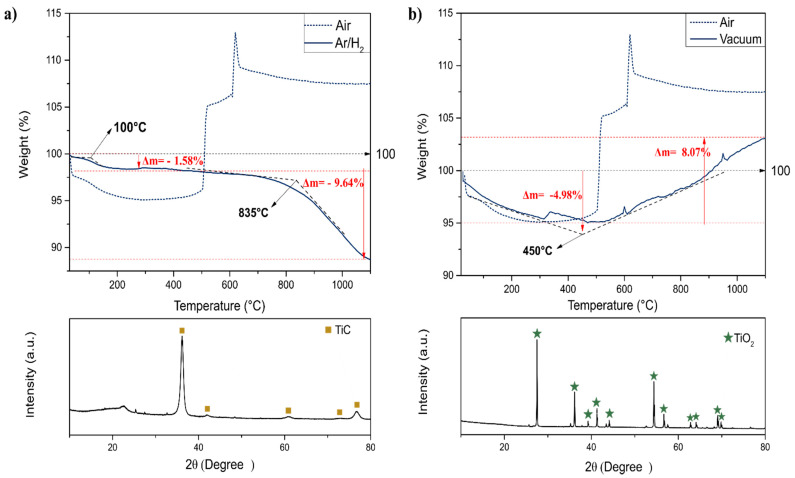
TG curves of T_3_C_2_T_x_ MXene (**a**) under vacuum and (**b**) in H_2_/Ar atmosphere and their respective XRD patterns representing the phases which were formed after TG measurement.

**Figure 11 membranes-12-01025-f011:**
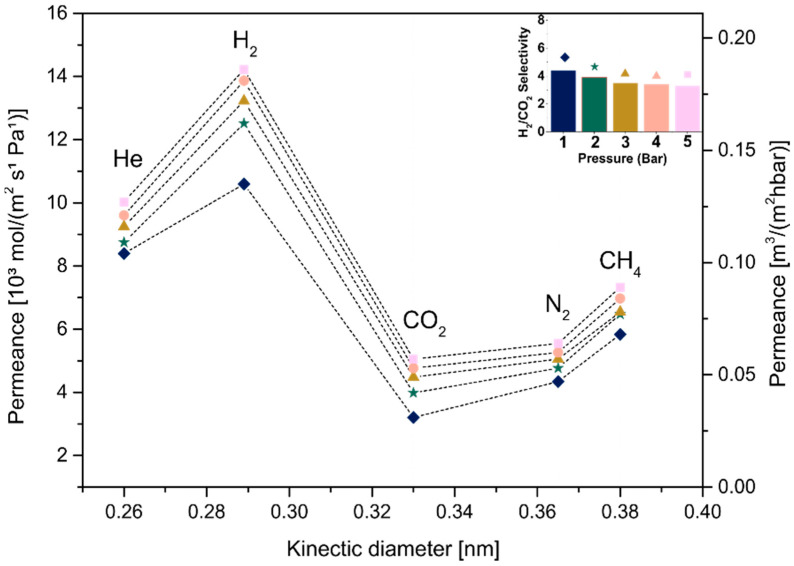
Single-gas permeabilities, permeances and H_2_/CO_2_ selectivity under different pressures for C_1__80.

**Figure 12 membranes-12-01025-f012:**
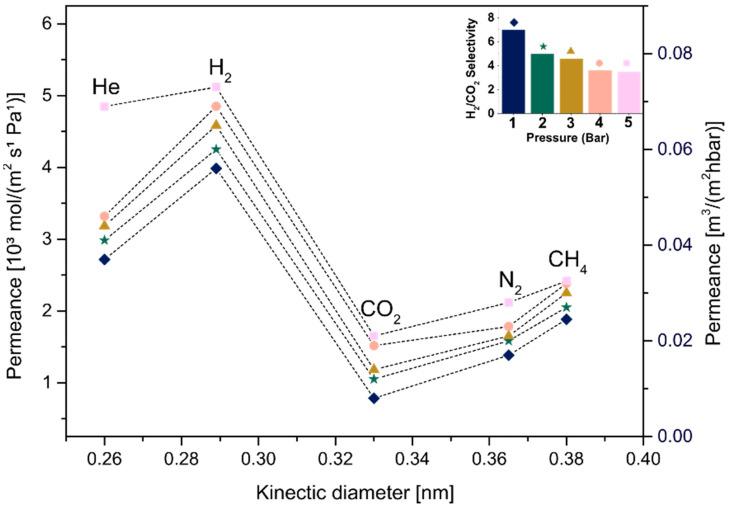
Single-gas permeabilities, permeances and H_2_/CO_2_ selectivity under different pressures for C_1__500.

**Figure 13 membranes-12-01025-f013:**
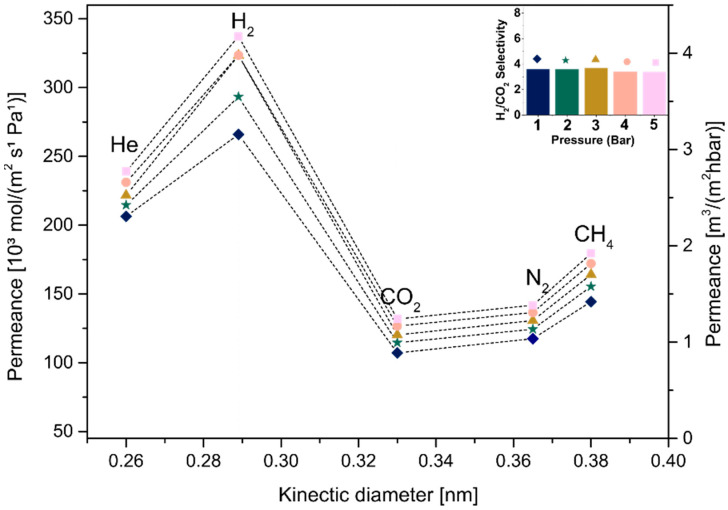
Single-gas permeabilities, permeances and H_2_/CO_2_ selectivity under different pressures for C_2__80.

**Figure 14 membranes-12-01025-f014:**
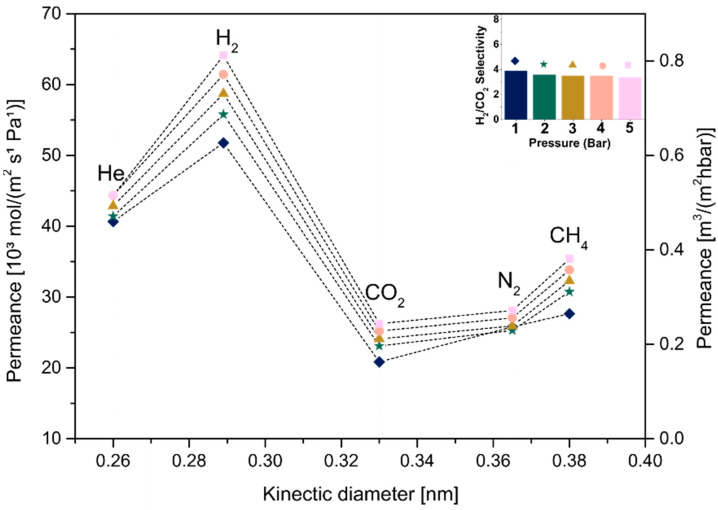
Single-gas permeabilities, permeances and H_2_/CO_2_ selectivity under different pressures for C_2__500.

**Table 1 membranes-12-01025-t001:** C-lattice constants and respective d-spacing obtained through Bragg’s law for samples produced with the most concentrated suspension (C_1_) and the least concentrated suspension (C_2_).

Sample ID	c-Lattice (Å)	d(002) (Å)	Free d-Spacing (nm)	Average Free D-Spacing (nm)
** *Heat treatment temperature 80 °C* **				
C_1__1	26.623	13.311	0.331	0.347 ± 0.003
C_1__2	27.036	13.518	0.352	
C_1__3	26.940	13.470	0.347	
C_1__4	26.782	13.391	0.339	
C_1__5	26.952	13.476	0.348	
C_2__1	27.079	13.540	0.354	0.352 ± 0.003
C_2__2	27.195	13.597	0.360	
C_2__3	26.755	13.378	0.338	
C_2__4	26.980	13.490	0.349	
C_2__5	27.092	13.546	0.355	
** *Heat treatment temperature 500 °C* **				
C_1__1	20.467	10.234	0.023	0.024 ± 0.001
C_1__2	20.413	10.206	0.021	
C_1__3	20.491	10.246	0.025	
C_1__4	20.580	10.290	0.029	
C_1__5	20.475	10.237	0.024	
C_2__1	20.773	10.387	0.039	0.040 ± 0.001
C_2__2	20.891	10.445	0.045	
C_2__3	20.808	10.404	0.040	
C_2__4	20.766	10.383	0.038	
C_2__5	20.903	10.452	0.045	

**Table 2 membranes-12-01025-t002:** H_2_ and CO_2_ permeabilities and permeances and the respective H_2_/CO_2_ selectivity.

Sample ID	H_2_ Permeability (Barrer)	H_2_ Permeance (10³ mol/m² s¹ Pa¹)	CO_2_ Permeability (Barrer)	CO_2_ Permeance (10³ mol/m² s¹ Pa¹)	H_2_/CO_2_ Selectivity
** *Heat treatment temperature 80 °C* **					
C_1__1	6.4	33.7	2.0	10.2	3.3
C_1__2	2.1	10.9	0.5	2.5	4.4
C_1__3	2.0	10.5	0.7	3.6	3.0
C_2__1	16.4	254.7	4.9	71.7	3.3
C_2__2	23.9	85.4	4.9	25.8	4.8
C_2__3	48.8	124.9	13.7	25.8	3.6
** *Heat treatment temperature 500 °C* **					
C_1__1	0.86	4.5	0.12	0.6	7.0
C_1__2	0.54	2.8	0.09	0.5	5.8
C_1__3	0.83	4.4	0.14	0.7	6.0
C_2__1	35.6	186.2	9.1	47.6	3.9
C_2__2	9.7	50.8	2.5	12.9	3.9
C_2__3	28.5	149.1	7.3	37.9	3.9

## Data Availability

Not applicable.

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
