# Peer review of "Ti3C2 MXene Membranes for Gas Separation: Influence of Heat Treatment Conditions on D-Spacing and Surface Functionalization"

_membranes, 2022, doi:10.3390/membranes12101025_

Round 1

Reviewer 1 Report

In this paper, the effects of heat treatment on the d- spacing of MXene nanosheets and on the surface functionalization of nanolayers was shown regarding its consequence on gas diffusion mechanism. The distance of the layers was reduced and defects were reduced and surface functionalization was maintained upon treatment of the Ti3C2 membrane at 500°C under Ar/H2 atmosphere as compared to the control one. This paper is well organized, and meet the journal requirements. The authors should consider the following concerns before publication. 

1. Some format errors happened in line 191, please correct this.

2. A graphical abstract figure is perferrable for this work.

3. How about the long term chemical stability of Ti3C2 flakes after heat treatment?

Author Response

Response to Reviewer 1 Comments

Point 1: Some format errors happened in line 191, please correct this.

Response 1: Thank you for the comment. The correction on line 191 has been made. After adding the suggested discussion and the graphical abstract, line 191 has been changed to a new position (line 206).

Point 2: A graphical abstract figure is preferable for this work.

Response 2: Thank you for the recommendation. The graphical abstract will certainly facilitate the understanding of this work. The graphical abstract has been added to the manuscript between lines 18 and 19.

Point 3: How about the long-term chemical stability of Ti3C2 flakes after heat treatment?

Response 3: Thank you very much for your question. After heat treatment, MXene membranes remain chemically stable due to the removal of water between adjacent nanosheets. Due its limitations to be used in humid environments and at higher temperatures, an oxygen-free atmosphere is required to avoid MXene flakes oxidation. Moreover, we need to consider that the more delaminated the MXene is the more exposed the surface will be, which may favor a faster oxidation process. Considering the decrease in d-spacing after heat treatment at high temperatures, stacked MXene favors a higher chemical stability when compared to individual flakes with a high free surface area, which facilitates oxidation. This discussion has been added in the manuscript and can be found in the lines 369-373 and 527-530.

Reviewer 2 Report

The paper has focused on Ti3C2 2D material in the application of gas separation as membrane, which is a hot topic in recent year and has been attracting much research interests. The authors have studied the effect of heat treatment on the d-spacing of Ti3C2 MXene. The resent work is comprehensive and the results are convincing and potentially useful to both the membrane and nanomaterial communities. I recommend publication in the journal of Membranes. However, there are still some limitations regarding the manuscript listed below.

First of all, this paper has focused on the d-spacing of Ti3C2 MXene. Do terminal functional groups also affect the selectivity performance of corresponding gases? As we know, the terminal functional groups of MXene may be desorbed or transformed under specific temperature or atmosphere conditions. This article also said that the removal of functional groups was observed, and the composition did not change. Is the change of interlayer spacing related to the removal of functional groups? Can functional groups change the interaction between MXene and gas components, thus affecting screening performance? I suggest the authors add some discussions regarding this issue.

Another question, how can we judge that the interaction between MXene layers becomes smaller? Will the interaction between layers become smaller when the layer spacing becomes smaller?

There are still some minor problems in the English of the article, which need to be expressed more smoothly and professionally.

Author Response

Point 1: First of all, this paper has focused on the d-spacing of Ti3C2 MXene. Do terminal functional groups also affect the selectivity performance of corresponding gases?

Response 1: Thank you for the pertinent question. These discussions will certainly strengthen our work. Besides the d-spacing, in case of molecular sieving, and molecular density factor, for Knudsen diffusion, the presence of the functional groups on the surface of MXene nanosheets also contributes to membrane separation performance. A few works have reported the adsorption capacity of CO2 molecule is related to its larger quadripolar moment in comparison to others gases such as N2, so then it influences the CO2 diffusion by its suppression through the membrane path. This effect leads to lower CO2 diffusion and it raises H2/CO2 selectivity. This discussion has been added in the manuscript and can be found in the line 95-104 and 487-491.

Point 2: As we know, the terminal functional groups of MXene may be desorbed or transformed under specific temperature or atmosphere conditions. This article also said that the removal of functional groups was observed, and the composition did not change. Is the change of interlayer spacing related to the removal of functional groups?

Response 2: Thank you for pointing this discussion out. Yes, the change in interlayer spacing can be related to the removal of both functional groups and water. In our work we mainly observed water removal and a slight removal of functional groups, without major changes in their nature and variety after heat treatment at higher temperatures, as shown in the peaks obtained through Raman spectroscopy. Even though there is partial removal of functional groups, we suppose that the decrease in defects and interlayer spacing was mainly associated with water removal and functional group modification at high temperature, e.g. -OH tends to change to -O, which contributed to a greater interaction between adjacent nanosheets. This discussion has been added in the manuscript and can be found in the line 362-365 and 483-487.

Point 3: Can functional groups change the interaction between MXene and gas components, thus affecting screening performance? I suggest the authors add some discussions regarding this issue.

Response 3: Thank you for your suggestions. Yes, functional groups can modify the interaction between MXene layers and diffuse gases, which can directly affect screening performance. The hydrophilic nature of the MXene surface provides a strong interaction between polar gases such as N2, CO2, NH3. As discussed in Point 1, CO2 can strongly interact with MXene functional groups during the diffusion mechanism due to its high quadrupole moment, providing efficient membrane gas separation performance. This discussion has been added in the manuscript and can be found in the line 487-491.

Point 4: Another question, how can we judge that the interaction between MXene layers becomes smaller? Will the interaction between layers become smaller when the layer spacing becomes smaller?

Response 4: Thank you again for your pertinent comment. We can assume that after heat treatment at higher temperature, the partial removal of the functional groups led to less interaction between the surface of the MXene nanosheet. However, since the removal happened in a smaller proportion, a considerable part of the functional groups as well as their variety was maintained, which contributed to the maintenance of a strong interaction between the MXene nanosheets. Moreover, the interaction between adjacent nanosheets favored change of functional groups at higher temperatures, i.e. from -OH to -O terminations, strengthening the bonds between adjacent layers. This discussion has been added in the manuscript and can be found in the line 321-326, 362-365 and 517-527.

Point 5: There are still some minor problems in the English of the article, which need to be expressed more smoothly and professionally.

Response 5: Thank you for your recommendation. A proper editing in English has been done to ensure a more professional language of our work.
